# Machine learning-based prediction of rheumatoid arthritis with development of ACPA autoantibodies in the presence of non-HLA genes polymorphisms

Grzegorz Dudek [1,2,3]*, Sebastian Sakowski[2,3], Olga Brzezińska[4], Joanna Sarnik[4], Tomasz Budlewski[4], Grzegorz Dragan[5], Marta Poplawska[6], Tomasz Poplawski[7], Michał Bijak[8], Joanna Makowska[4]*

1 Electrical Engineering Faculty, Czestochowa University of Technology, Czestochowa, Poland, 2 Faculty of Mathematics and Computer Science, University of Lodz, Lodz, Poland, 3 Centre for Data Analysis, Modelling and Computational Sciences, University of Lodz, Lodz, Poland, 4 Department of Rheumatology, Medical University of Lodz, Lodz, Poland, 5 Department of Clinical Chemistry and Biochemistry, Medical University of Lodz, Lodz, Poland, 6 Biobank, Department of Immunology and Allergy, Medical University of Lodz, Lodz, Poland, 7 Department of Pharmaceutical Microbiology and Biochemistry, Medical University of Lodz, Lodz, Poland, 8 Biohazard Prevention Centre, Faculty of Biology and Environmental Protection, University of Lodz, Lodz, Poland

* grzegorz.dudek@pcz.pl (GD); joanna.makowska@umed.lodz.pl (JM)

**Data Availability Statement:** All data files are available from the github repository, https://github.com/GMDudek/ML-RA.

## Abstract

Machine learning (ML) algorithms can handle complex genomic data and identify predictive patterns that may not be apparent through traditional statistical methods. They become popular tools for medical applications including prediction, diagnosis or treatment of complex diseases like rheumatoid arthritis (RA). RA is an autoimmune disease in which genetic factors play a major role. Among the most important genetic factors predisposing to the development of this disease and serving as genetic markers are HLA-DRB and non-HLA genes single nucleotide polymorphisms (SNPs). Another marker of RA is the presence of anticitrullinated peptide antibodies (ACPA) which is correlated with severity of RA. We use genetic data of SNPs in four non-HLA genes (PTPN22, STAT4, TRAF1, CD40 and PADI4) to predict the occurrence of ACPA positive RA in the Polish population. This work is a comprehensive comparative analysis, wherein we assess and juxtapose various ML classifiers. Our evaluation encompasses a range of models, including logistic regression, *k*-nearest neighbors, naïve Bayes, decision tree, boosted trees, multilayer perceptron, and support vector machines. The top-performing models demonstrated closely matched levels of accuracy, each distinguished by its particular strengths. Among these, we highly recommend the use of a decision tree as the foremost choice, given its exceptional performance and interpretability. The sensitivity and specificity of the ML models is about 70% that are satisfying. In addition, we introduce a novel feature importance estimation method characterized by its transparent interpretability and global optimality. This method allows us to thoroughly explore all conceivable combinations of polymorphisms, enabling us to pinpoint those possessing the highest predictive power. Taken together, these findings suggest that non-HLA

**Funding:** National Science Center (NCN, Poland), grant number UMO-2017/25/B/NZ6/01358. The funders (National Science Center) had no role in study design, data collection and analysis, decision to publish, or preparation of the manuscript.

**Competing interests:** The authors have declared that no competing interests exist.

SNPs allow to determine the group of individuals more prone to develop RA rheumatoid arthritis and further implement more precise preventive approach.

## 1 Introduction

Artificial Intelligence (AI), particularly Machine Learning (ML), have become increasingly popular tools for medical applications including prediction of diseases, diagnosis or treatment based on presence of specific markers. Application of ML can be useful for analysis autoimmunology diseases like rheumatoid arthritis (RA).

RA is a complex, chronic disease with a range of symptoms and underlying factors, making it difficult to accurately diagnose or predict. Interplay between genetic, environmental factors and autoimmunity triggers [1] is supposed to lead to the development of inflammation that begins in the synovial membrane of the joints and then spreads to other tissues, leading to systemic inflammation in patients with RA. This process is preceded by the appearance of RA-specific antibodies directed against various antigens including anticitrullinated peptide. The presence of anticitrullinated peptide antibodies (ACPA) is one of the factors that not only precedes the onset of RA, but also leads to the more aggressive course of the disease including joint damage, increased disease activity, and worse functional outcomes [2].

The most common genetic risk factors for the development of RA are certain variants in HLA–DRB1 loci called shared epitopes [3]. Also single nucleotide polymorphisms (SNPs) in non-HLA genes (PTPN22, TRAF1, STAT4, PADI4 and CD40) are linked to an increased risk of development of RA. PTPN22 encodes the protein tyrosine phosphatase non-receptor type 22 that plays a crucial role in the regulation of T cell activation and signaling. TRAF1 and STAT 4 are responsible for immune cell activation [4] whereas PADI4 is responsible for converting arginine to citrulline and forming citrullinated peptides being autoantigens. The CD40 gene encodes a protein responsible for the regulation of the immune response and communication between various immune cells. Genetic variation in two of these five genes has been linked with ACPA RA [4, 5].

### 1.1 Related work

The multifactorial nature of the RA is one of the reasons why ML and AI are suitable tools for the analysis of diagnosis and treatment of RA. ML models can help identify patterns in patient data, including genetic data that can be difficult for humans to discern, leading to more accurate and early diagnosis. ML can help predict the progression of the disease, allowing clinicians to personalize treatment plans based on the specific needs of a patient.

There are many studies that demonstrate the effectiveness of ML in the diagnosis and prognosis of RA. For example, the ML model described in [6] shows promise in guiding treatment decisions in clinical practice, based primarily on clinical profiles with additional genetic information. In this work, Gaussian process regression effectively remapped the feature space and identified subpopulations that do not respond well to anti-TNF treatments. In [7], ML methods such as logistic regression, random forest, support vector machine, gradient tree boosting, and extreme gradient boosting are used for genomic prediction of RA and systemic erythematous lupus. They were able to differentiate these two diseases with robust performance based on genetic variations in HLA-DQA1, HLA-DQB1, and HLA-DRB1.

ML models developed in [8] allowed predicting the response of patients with RA to tumor necrosis factor inhibitors exclusively using data available in the clinical routine. The authors compared the performances of multiple models such as linear regression, random forest,

XGBoost and CatBoost. In [9], ML methods were considered feasible to predict flares after reducing disease modifying antirheumatic drugs in patients with RA in sustained remission. Four basic ML models were trained (logistic regression, k-nearest neighbors, naïve Bayes classifier, and random forests), and their predictions were additionally combined to train an ensemble learning method, a stacking meta-classifier model to predict the individual flare probability. The current status and future perspectives of using AI and ML in RA based on a detailed review of the literature are presented in [10].

## 1.2 Contribution

This work explores the association of five SNPs in non-HLA genes PADI4 (rs2240340), STAT4 (rs7574865), CD40 (rs4810485), PTPN22 (rs2476601), and TRAF1 (rs3761847) with RA development. The results of this study will greatly contribute to better prediction the occurrence of ACPA positive RA in the Polish population.

This is the first work evaluating the feasibility of ML models in predicting ACPA-positive RA. It makes significant contributions to the current research in several ways:

1. Bridging a Gap in Genetic Markers for RA: The research investigates specific polymorphisms within non-HLA genes (PTPN22, STAT4, TRAF1, CD40, PADI4) to predict the aggressive progression of RA. This is crucial as identifying these genetic markers can lead to early detection and more personalized treatment strategies.

2. Evaluating ML Algorithms for RA Prediction: Various ML algorithms are assessed and compared for their effectiveness in predicting RA based on genetic data. The algorithms include logistic regression, $k$-nearest neighbors, naïve Bayes, decision trees, boosted trees, multilayer perceptrons, and support vector machines. This comparative analysis aims to determine which ML models are most effective for this purpose.

3. Feature Importance and Optimal Polymorphism Combinations: The study introduces a novel feature importance estimation method that enables the exploration of all possible combinations of polymorphisms. This approach aims to identify those with the highest predictive accuracy for RA, providing a clear and globally optimal interpretation of the genetic data.

The subsequent sections of this work are structured as follows. Section 2 briefly describes the medical and genetic methodology. In Section 3, we provide an in-depth exploration of the ML models employed in our study. Section 4 combines the experimental study with a comprehensive discussion. Finally, Section 5 summarizes the key findings of our research and presents suggestions for future directions.

## 2 Patients and methods

### 2.1 Patients

The study group included 78 patients with RA anti-citrullinated antibodies positive (ACPA+) selected from patients of the Department of Rheumatology, Medical University of Lodz and the outpatient clinic. All patients were diagnosed with RA according to EULAR/ACR 2010 diagnostic. The control group of 78 volunteers was recruited from patients without any symptoms of chronic inflammatory condition who were consulted in outpatients clinic or hospitalized, as well as from a group of potential marrow donors.

This cohort study was approved by the Institutional Bioethics Committee of the Medical University of Lodz (no. RNN/07/18/KE, approved date: 16 January 2018; Participant consent: written; The study did not include minors; We do not report a retrospective study of medical

**Table 1. Clinical characteristic of the studied group.**

|  | RA PATIENTS | CONTROL |
|---|---|---|
|  | *N* = 78 | *N* = 78 |
| Sex | F 63; M15 | F 50; M28 |
| AGE [years] | 60.00±13.08 | 35.72±10.55 |
| Disease duration [years] | 10.65±10.23 | - |
| Remission [DAS28<2.6] | yes 11; no 60; no data 7 | - |
| CRP [mg/l] | 16.12±22.76 | - |
| ERS [MM/H] | 26.50±23.03 | - |
| RF [IU/ml] | 189.50±360.40 | - |
| ACPA [RU/ML] | 215.57±398.65 | - |
| Treatment | Methotrexate 34/78 | - |
|  | Sulfasalazine 4/78 |  |
|  | Leflunomide 10/78 |  |
|  | Hydroxychloroquine 2/78 |  |
|  | GCS 43/78 |  |

where: CRP—c-reactive protein; ESR—eosinophil sedimentation rate; RF—rheumatoid factor; ACPA—anti-citrullinated peptide antibodies.

records or archived samples). The study group included 78 patients with RA (63 women and 15 men; mean age 60.00±13.08 years). Mean disease duration time was 10.65±10.23 years. All patients were anti-citrullinated peptide antibodies positive (mean concentration 215.57 ±398.65 RU/ml) and 88.46% were rheumatoid factor positive (mean concentration 189.50 ±360.40 IU/ml). Level of inflammatory markers (CRP 16.12±22.76 mg/l; ERS 26.50±23.03 mm/h) were determined. Two thirds of patients during the qualification for the study was treated with disease modifying drug (methotrexate 43.6sulfasalazine 5.1%; leflunomide 12.8%; hydroxychloroquine 2.6%) in addition 55.3% took glucocorticosteroids. The control group included 78 volunteers without any autoimmunological and inflammatory diseases. This group included 50 women and 28 men in mean age 35.72±10.55 years. Table 1 characterizes both groups.

## 2.2 Genotyping

Genomic DNA (gDNA) was isolated from peripheral blood (K3EDTA tubes) collected from patients and controls by using GeneMatrix Blood DNA purification Kit (EURx) according to the manufacturer protocol. Genotypes were determined by Taqman SNP Genotyping Assay (Thermo Fisher Scienific) and HOT FIREPol® Probe qPCR Mix (Solis). Analysis was made in Bio-Rad CFX96 system (BioRad) according to the manufacturer protocol. The analyzed SNPs are presented in Table 2. They are treated as features (inputs) in the ML models.

## 3 Machine learning models

From ML point of view, the problem is to classify patients and controls into one of two classes (RA/healthy) based on v1-v5 SNPs in non-HLA genes. In the preliminary experiments, we tested several ML models to select the most accurate ones for further study. They included: logistic regression (LR), *k*-nearest neighbors (kNN), naïve Bayes (NB), decision tree (DT), boosted trees (BT), multilayer perceptron (MLP), and support vector machine (SVM). Note that kNN, NB, DT, and BT can work with nominal features such as our genetic data (see v1-v5 in Table 2) but other models work only with numerical data. For them, nominal values were

**Table 2. Basic information of the five selected SNPs in non-HLA genes.**

| Symbol | Gene.SNP[1] | Chr[2] | Positions[3] | Allele | Assay ID[4] |
|---|---|---|---|---|---|
| v1 | PADI4.rs2240340 | 1 | 17336144 | C/T | C__16176717_10 |
| v2 | TRAF1.rs3761847 | 9 | 120927961 | A/G | C___2783640_10 |
| v3 | STAT4.rs7574865 | 2 | 191099907 | G/T | C__29882391_10 |
| v4 | CD40.rs4810485 | 20 | 46119308 | G/T | C___1260190_10 |
| v5 | PTPN22.rs2476601 | 1 | 113834946 | A/G | C__16021387_20 |

[1] according to dbSNP database;

[2] chromosome;

[3] chromosome position according to the Genome Reference Consortium Human Build 38;

[4] related to Thermo Fisher Scienific.

converted to numeric dummy variables. Since each feature v1-v5 takes one of three values, the binary variables encoding each of them are 3-bit long.

Based on the results reported in Section 3, the four most accurate models were selected: NB, DT, BT, and SVM. These models operate on different principles and techniques, with each having its own strengths and weaknesses. They are characterized below. The following notations were adopted:

- $\mathbf{x} = [v_1, \ldots, v_5]$ denotes a sample (patient),

- $y \in \{-1, 1\}$ represents a class: healthy ($-1$) or RA ($1$),

- $N$ is a number of samples (156 in our case) and

- $n$ is a number of features (5 in our case).

## 3.1 Naïve bayes

NB classifier is a probabilistic algorithm that calculates the probability of each class based on the input features. It then assigns the sample to the class with the highest probability. It assumes that all features are independent of each other. The NB algorithm is computationally efficient and can handle a large number of predictors, making it a popular algorithm in many applications.

In our case, for nominal features and two classes, the NB decision rule (maximum a posteriori decision rule) is as follows:

$$\hat{y} = \underset{y \in \{-1,1\}}{\operatorname{argmax}} \, P(y) \prod_{j=1}^{n} P(v_j | y) \qquad (1)$$

where $P(y)$ is the prior probability that a class index is $y$ and $P(v_j|y)$ is a conditional probability of $v_j$ given class $y$.

In the NB classifier, each feature/class combination is treated as a separate, independent multinomial random variable. This allows us to estimate the complex multivariate distribution $P(\mathbf{x}|y)$ using the product of univariate distributions $P(v_j|y)$, due to the "naive" assumption of conditional independence between every pair of features given the value of the class variable. This simplification is helpful in addressing the curse of dimensionality and reduces the computational complexity of the model, making it faster to train and apply. Despite its

oversimplified assumptions, the NB classifier has been successful in many real-world scenarios. However, its performance may be affected by the quality of the data and the presence of irrelevant or correlated features, among other factors.

## 3.2 Decision tree

DT classifier is a non-parametric algorithm that partitions the feature space recursively into smaller and smaller regions by selecting the feature that provides the most information gain at each node. It is interpretable and can handle both numerical and categorical data.

We use the CART (classification and regression tree) implementation of DT [11]. DT construction is a process of recursively partitioning a dataset into subsets, based on the values of the features. It involves selecting the best feature to split the data at each node based on some impurity measure, such as Gini index or entropy, and then repeating the process recursively until some stopping criterion is met, such as reaching a predefined maximal number of splits or achieving a minimum number of samples in each leaf node. The resulting tree can then be used to make predictions for new samples by traversing the tree from the root node to one of the leaf nodes that corresponds to the predicted class. The tree can be interpreted as a set of simple rules that can be easily understood by humans.

The decision rule of DT can be expressed as follows:

$$\hat{y} = \sum_{l \in L} label(l) I(\mathbf{x} \in l) \tag{2}$$

where $L$ is a set of leaves, $label(l)$ is a function, which assigns a label to leaf $l$ based on the subset of samples that reached that leaf (typically the label is a majority class), and $I(\mathbf{x} \in l)$ returns 1 when sample $\mathbf{x}$ reaches leaf $l$ and 0 otherwise.

DT is a powerful yet conceptually simple approach. It partitions the feature space into a set of rectangles and fits a simple model in each one. In our case, this model is represented by function *label*. A single tree fully describes the partition of the feature space. The decision boundary in DT is composed of segments that are parallel to the coordinate axes, which characterizes this method. The tree representation is popular among medical scientists, possibly because it emulates the way a doctor thinks.

## 3.3 Boosted trees

There are several variations of boosted DTs, each with its own set of characteristics. In our study, we made a deliberate choice of algorithm, guided by the preliminary optimization procedure outlined in Section 4.1. Our selection led us to the Gentle Adaptive Boosting (GAB) algorithm, as proposed in [12].

In GAB, a base learner (DT) is iteratively trained on the data, with each subsequent iteration focusing more on the samples that were misclassified by the previous models. The algorithm assigns higher weights to these samples, so the subsequent base learners are more likely to correctly classify them. GAB is a powerful and robust algorithm for ensemble learning, with a soft weighting scheme that can lead to more stable learning in noisy or overlapping datasets.

GAB starts with equal weights ($w = 1/N$) for all training samples. At the successive $M$ steps, it fits regression models $f_m(\mathbf{x})$, $m = 1, .., M$ (regression trees in our case) by weighted least-squares of $y$ to $\mathbf{x}$. The minimized error at step $m$ is $\sum_{i=1}^{N} w_i^m (y_i - f_m(\mathbf{x}_i))^2$, where $w_i^m$ is a weight of the $i$-th sample at the $m$-th step. The weights are updated as follows: $w_i^{m+1} = w_i^m \exp(-y_i f_m(\mathbf{x}))$ and then normalized. The final decision of tha GAB classifier is as follows

[12]:

$$\hat{y} = \text{sign}\left(\sum_{m=1}^{M} f_m(\mathbf{x})\right) = \text{sign}\left(\sum_{m=1}^{M} \sum_{l \in L_m} label(l) I(\mathbf{x} \in l)\right) \tag{3}$$

where $L_m$ is a set of leaves of the $m$-th tree.

### 3.4 Support vector machine

SVM is a linear classifier that finds the hyperplane that maximizes the margin between the classes. It can also handle nonlinear data by transforming the data into a higher-dimensional space using kernel functions. It is powerful and performs well on a wide range of classification tasks.

The decision rule of SVM is expressed as follows:

$$\hat{y} = \text{sign}\left(\sum_{i=1}^{N} \alpha_i y_i k(\mathbf{x}_i, \mathbf{x}) + b\right) \tag{4}$$

where $\alpha_i \in [0, C]$ are Lagrange multipliers which are positive for support vectors and zero for other training samples, $b$ is a parameter, and $k(\mathbf{x}_i, \mathbf{x})$ is a kernel function, e.g. Gaussian-type kernel $k(\mathbf{x}_i, \mathbf{x}) = \exp\left(-\frac{\|\mathbf{x}_i - \mathbf{x}\|^2}{\sigma^2}\right)$.

Parameter $C$, also known as the box constraint, is the penalty/regularization parameter of the error term. It plays a key role in the SVM algorithm because it controls the trade-off between maximizing the margin and minimizing the classification error on the training data. A higher value of $C$ results in a narrower margin and fewer misclassifications, which may lead to overfitting. The second key hyperparameter is a kernel width parameter, $\sigma$ in Gaussian kernel. It determines the shape of the decision boundary. A lower value of $\sigma$ leads to a more complex decision boundary and overfitting.

## 4 Experimental study

In this section, we evaluate the effectiveness of our proposed ML models in predicting RA based on non-HLA gene polymorphisms. Initially, we compare the performance of seven different ML models and select the top four performing ones for further testing. Finally, we identify the most important features that contribute to the prediction of RA using these models.

To gain insight into the input genetic data, we plotted the empirical distributions of the features in both classes, RA and healthy, as shown in Fig 1. These distributions represent the conditional probabilities $P(v_j|y)$, which are used by the NB classifier to determine the sample class, as described in Eq (1).

### 4.1 Optimisation, training and evaluation setup

To optimize the hyperparameters of the ML models, we employed Bayesian optimization combined with 5-fold cross-validation. This approach allows for efficient exploration of the hyperparameter space and helps to avoid overfitting on the training data. The hyperparameter exploration ranges were as follows:

- LR has no hyperparameters.

- kNN: Number of neighbors $k$ was searched among integers log-scaled in the range $[1, N/2]$; Distance metric: due to nominal features, the Hamming distance was used.

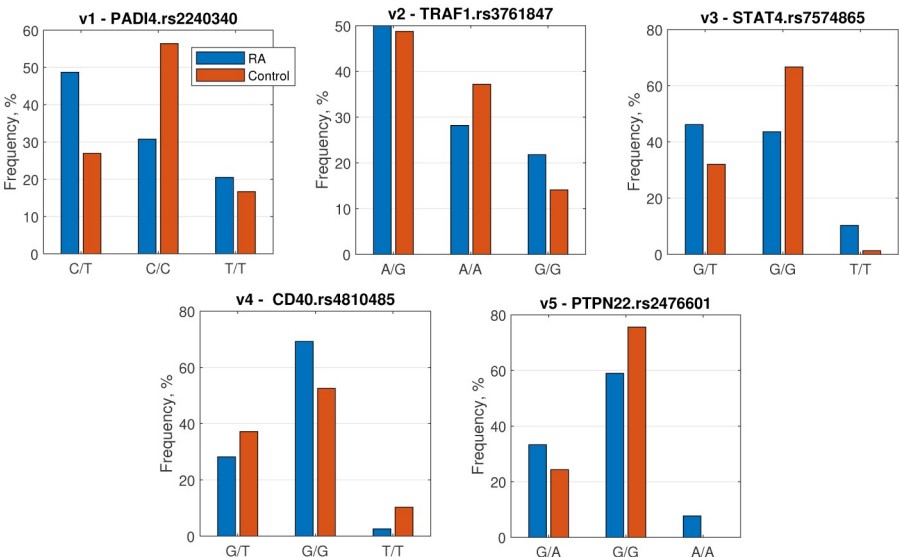

**Fig 1. Association of RA with the frequency of SNPs in PTPN22 (rs2476601), PADI4 (rs2240340), TRAF1 (rs3761847), STAT4 (rs7574865), and CD40 (rs4810485) genes.**

- NB: Due to nominal features, a variant of NB with multivariate multinomial distribution was selected, where each predictor/class combination is a separate, independent multinomial random variable.

- DT: Maximum number of splits was searched among integers log-scaled in the range $[1, N − 1]$; Split criterion was searched among Gini's diversity index and cross entropy.

- BT: Ensemble method was searched among AdaBoost, LogitBoost (adaptive logistic regression), GAB, and Random Forest; Maximum number of splits was searched among integers log-scaled in the range $[1, N − 1]$; Number of learners was searched among integers log-scaled in the range $[10, 500]$; Learning rate was searched among real values log-scaled in the range $[0.001, 1]$; Number of features to sample was searched among integers in the range $[1, n]$, where $n$ is the number of features.

- MLP: Number of fully connected layers was searched among 1, 2, and 3 layers; Layer size was searched among integers log-scaled in the range $[1, 300]$; Activation was searched among ReLU, Tanh, None, and Sigmoid; Regularization strength was searched among real values log-scaled in the range $[0.00001/n, 100000/n]$.

- SVM: Kernel function was searched among Gaussian, Linear, Quadratic, and Cubic; Box constraint level was searched among positive values log-scaled in the range $[0.001, 1000]$; Width parameter was searched among positive values log-scaled in the range $[0.001, 1000]$.

The optimal values of hyperparameters are shown in Table 3.

The ML models were evaluated using leave-one-out cross-validation. In the preliminary tests, we compare all the models and select the most accurate. Then, we evaluate these selected

**Table 3. Hyperparameters of the ML models.**

| Model | Hyperparameters |
|-------|-----------------|
| LR | - |
| kNN | $k = 1$, distance metric: Hamming |
| NB | A variant for conditional multivariate multinomial distribution for features was selected |
| DT | Split criterion: cross entropy, maximum number of splits: 47 |
| BT | Ensemble method: GAB, maximum number of splits: 150, number of trees: $M = 10$, number of features to sample: 5, learning rate: 0.01 |
| MLP | Number of fully connected layers: 3, activation: ReLU, number of nodes in each layer: 10, iteration limit: 1000 |
| SVM | Kernel function: Gaussian, width parameter: $\sigma = 0.56$, Box constraint level: $C = 1$ |

models more precisely using the following metrics:

$$\text{Accuracy}: \quad Acc = (TP + TN)/N \tag{5}$$

$$\text{Precision}: \quad Prec = TP/(TP + FP) \tag{6}$$

$$\text{Sensitivity}: \quad Sens = TP/(TP + FN) \tag{7}$$

$$\text{Specificity}: \quad Spec = TN/(TN + FP) \tag{8}$$

$$\text{F1 score}: \quad F1 = (2TP)/(2TP + FP + FN) \tag{9}$$

where *TP*, *FP*, *FN* and *TN* denote true positive, false positive, false negative and true negative, respectively.

All algorithms were implemented in Matlab 2022b and run on a ten-core CPU (Intel i7-6950x, 3.0 GHz, 48 GB RAM, Windows 10 Pro).

## 4.2 Results

Table 4 shows accuracy of the models, their training times and predictive speeds. The table highlights that two models achieved both the highest accuracy and the lowest training time: NB and SVM. This indicates that these models are particularly efficient in their ability to accurately classify data while also requiring relatively little time to train. Slightly worse in terms of accuracy were BT, DT and MLP. A distance-based classifier, kNN, is clearly worse in terms of accuracy than other models. Taking into account the results from Table 4, we selected four models for further research: NB, DT, BT and SVM.

Table 5 presents a comparative analysis of model performances assessed through leave-one-out cross-validation. Notably, all models exhibit very similar accuracy. To rigorously examine

**Table 4. Results of preliminary experiments.**

| Model | LR | kNN | NB | DT | BT | MLP | SVM |
|-------|-----|-----|-----|-----|-----|-----|-----|
| Accuracy | 0.654 | 0.59 | 0.699 | 0.673 | 0.679 | 0.673 | 0.699 |
| Training time | 5.78 | 4.61 | 2.71 | 4.23 | 6.2 | 18.98 | 2.71 |
| Prediction speed | 640 | 520 | 700 | 920 | 360 | 810 | 1100 |

where: training time in seconds and prediction speed in obs/sec.

**Table 5. Comparison of ML model performances.**

| Model | TP | FP | FN | TN | Acc | Prec | Sens | Spec | F1 |
|---|---|---|---|---|---|---|---|---|---|
| NB | 52 | 22 | 26 | 56 | 0.6923 | 0.7027 | 0.6667 | 0.7179 | 0.6842 |
| DT | 56 | 25 | 22 | 53 | 0.6987 | 0.6914 | 0.7179 | 0.6795 | 0.7044 |
| BT | 53 | 23 | 25 | 55 | 0.6923 | 0.6974 | 0.6795 | 0.7051 | 0.6883 |
| SVM | 61 | 30 | 17 | 48 | 0.6987 | 0.6703 | 0.7821 | 0.6154 | 0.7219 |

the hypothesis that the predictions from each model have equivalent accuracy in predicting true class labels, we employ a mid-$p$-value McNemar test. This test, recommended by Dietterich [13], is particularly suitable when data is limited, and each algorithm can only be evaluated once. The results of the test affirm that, at the 1% significance level, all models yield statistically indistinguishable results.

Sensitivity and specificity stand as pivotal metrics in gauging the accuracy of a diagnostic tool, providing healthcare providers with insights into its effectiveness. In our findings, we achieved an approximate 70% sensitivity and specificity, indicating that this percentage of patients were correctly diagnosed (sensitivity), or controls were correctly identified with negative results (specificity).

DT is the most interpretable of the models tested, as it is shown in Fig 2. The extensive structure of the tree highlights the complexity of the task, as the model requires numerous decision rules to accurately classify the data. Despite its complexity, decision trees are often favored for their interpretability and transparency, as they allow for a clear understanding of the decision-making process used by the model.

**4.2.1 Imbalanced data.** Our dataset exhibits a balanced distribution, where the "RA" and "healthy" classes are equivalent in terms of sample count. However, it is crucial to recognize that imbalanced datasets can introduce bias into model performance assessments. Many ML algorithms are optimized to maximize overall accuracy, which can be misleading in scenarios where class distributions are skewed. In such cases, a model may achieve seemingly high

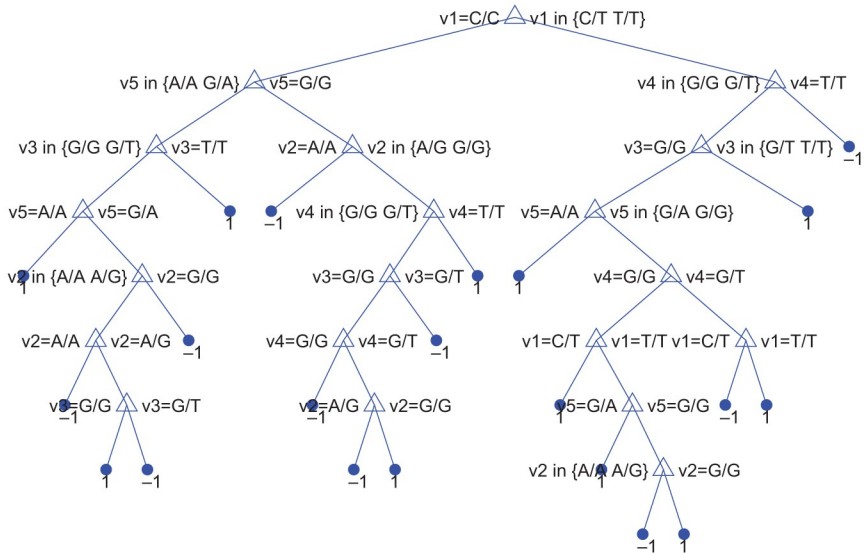

**Fig 2. DT model.**

**Table 6. Comparison of ML model performances for imbalanced data (imbalance ratio = 2).**

| Model | TP | FP | FN | TN | Acc | Prec | Sens | Spec | F1 |
|---|---|---|---|---|---|---|---|---|---|
| NB | 13.44 | 7.32 | 25.56 | 70.68 | 0.7190 | 0.6480 | 0.3446 | 0.9062 | 0.4469 |
| DT | 17.66 | 13.50 | 21.34 | 64.50 | 0.7022 | 0.5657 | 0.4528 | 0.8269 | 0.4999 |
| BT | 18.46 | 16.28 | 20.54 | 61.72 | 0.6853 | 0.5305 | 0.4733 | 0.7913 | 0.4992 |
| SVM | 7.32 | 10.46 | 31.68 | 67.54 | 0.6398 | 0.4015 | 0.1877 | 0.8659 | 0.2543 |

accuracy by predominantly predicting the majority class, while its performance on the minority class suffers.

To comprehensively evaluate our models' performance on imbalanced data, we consider two imbalance ratios. Specifically, we vary the ratio of "healthy" samples (the majority class) to "RA" samples (the minority class) to be either 2 or 10. For each configuration, we conduct 50 model training iterations, drawing either 39 or 8 samples from a pool of 78 "RA" samples, while keeping the "healthy" class constant.

Tables 6 and 7 present a comparative analysis of the results. The critical metrics are sensitivity, which reflects the accuracy of classifying "RA" samples, and specificity, which corresponds to the accuracy of classifying "healthy" samples.

As depicted in Table 6, when considering an imbalance ratio of 2, the overall accuracy remains relatively stable compared to the balanced scenario. However, the *Sens* metric reveals a significant decline in the recognition accuracy of the "RA" class. Among the algorithms we examined, BT exhibits the smallest reduction, approximately 30%, while SVM shows the most substantial decrease at 76%. When the imbalance ratio increases to 10 (as shown in Table 7), the degradation in accuracy for the "RA" class becomes much more pronounced, ranging from 78% for BT to a substantial 97% for SVM.

Our findings align with prior research, reinforcing the idea that ensemble methods like BT can effectively enhance performance on imbalanced data by combining multiple weak learners to form a strong learner. Such methods often outperform individual models, especially when dealing with highly skewed class distributions.

It is worth noting that DT emerges as the second-best performing model for imbalanced data. In the case of an imbalance ratio of 2, its quality metrics closely resemble those of BT. However, when the imbalance ratio increases to 10, DT's sensitivity is nearly half that of BT but still more than twice as high as those of NB and SVM.

**4.2.2 Feature importance.** To identify the most important features, we ran the models using all possible combinations of features, as shown in Fig 3. It is worth noting that in this study, we evaluate feature importance in the context of applied models and consider features not individually, but in groups. This approach allowed us to assess the interrelationships between features and their collective impact on classification accuracy. By evaluating features in this way, we can gain insights into which combinations of features are most useful for predicting the target variable, and use this information to optimize model performance.

**Table 7. Comparison of ML model performances for imbalanced data (imbalance ratio = 10).**

| Model | TP | FP | FN | TN | Acc | Prec | Sens | Spec | F1 |
|---|---|---|---|---|---|---|---|---|---|
| NB | 0.30 | 0.48 | 7.70 | 77.52 | 0.9049 | - | 0.0375 | 0.9938 | 0.0582 |
| DT | 0.64 | 1.90 | 7.36 | 76.10 | 0.8923 | 0.2252 | 0.0800 | 0.9756 | 0.1095 |
| BT | 1.20 | 4.18 | 6.80 | 73.82 | 0.8723 | 0.1977 | 0.1500 | 0.9464 | 0.1675 |
| SVM | 0.20 | 2.50 | 7.80 | 75.50 | 0.8802 | 0.0600 | 0.0250 | 0.9679 | 0.0347 |

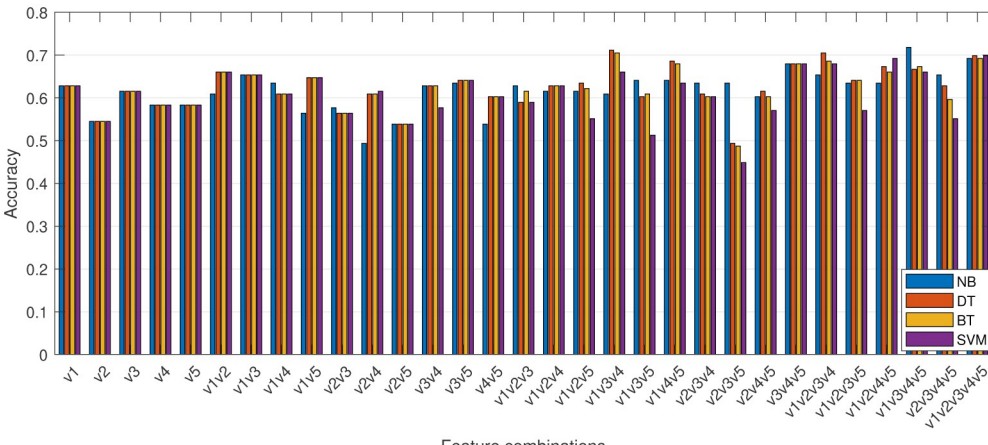

**Fig 3. Accuracy of the models for different combinations of input features.**

While using all features can generally result in high accuracy for all models, there are certain feature combinations that can improve the performance of some models beyond an accuracy of 0.7. For example, feature combination v1v3v4v5 was particularly effective for NB (this case is visualized with a confusion matrix in Fig 4), while v1v2v3v4 was found to be important for DT. Additionally, feature combination v1v3v4 was found to be useful for both DT and BT. It is

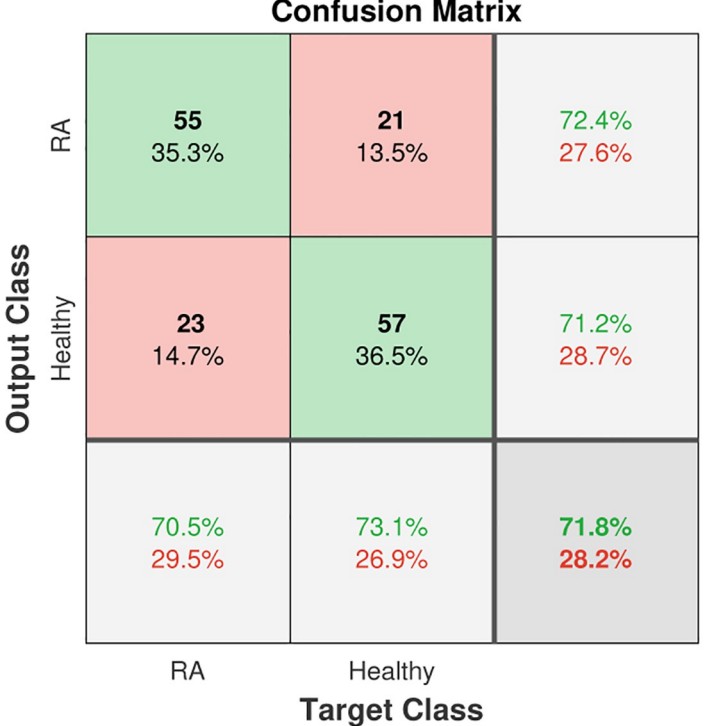

**Fig 4. Confusion matrix for NB with features v1v3v4v5.**

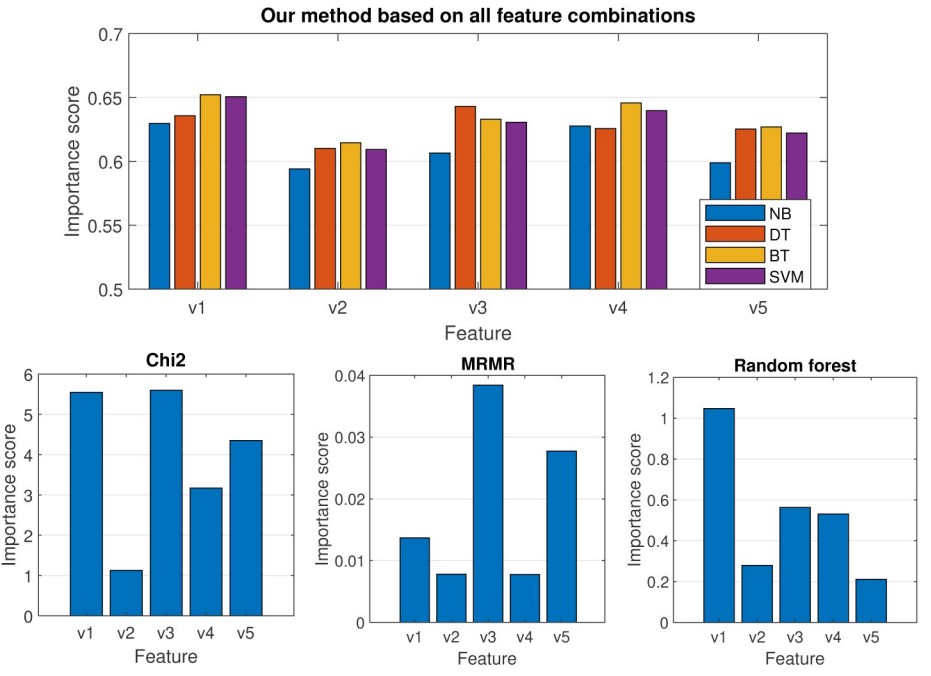

**Fig 5. Feature importance.**

important to note that the feature combination v2v3v5 resulted in the significantly lower accuracy, below 0.5, for all models except DT.

To evaluate the predictive power of individual features, we estimated their importance scores by calculating the average accuracy of the classifier across all feature combinations that include the respective feature. The results are presented in Fig 5. For comparison, we also included the results obtained using three popular feature importance estimation methods based on chi2, minimum redundancy maximum relevance algorithm (MRMR) and random forest. Chi2 is a statistical method that evaluates the independence between features and the target variable based on the chi-squared test. MRMR, on the other hand, finds an optimal set of features that is mutually and maximally dissimilar while effectively representing the response variable [14]. Feature importance estimation using random forest involves evaluating the contribution of each feature in the predictive accuracy of the random forest classifier. The importance of a feature is determined by measuring the decrease in the model's accuracy when that feature is randomly permuted or shuffled while keeping the rest of the data unchanged.

The feature importance scores obtained using different methods, as presented in Fig 5, reveal significant differences. Our method ranks the features in the following order of importance: v1, v4, v3, v5, and v2 (for DT the order is different: v3, v1, v4, v5, and v2). In contrast, the order given by chi2 is v3, v1, v5, v4, and v2, MRMR ranks the features as v3, v5, v1, and v2/ v4 (with v2 and v4 having the same score), while random forest ranks the features as v1, v3, v4, v2, and v5.

## 4.3 Discussion

Rheumatoid arthritis is a chronic disease of not fully elucidated pathogenesis in which both genetic background and environmental risk factors play role [1]. Creating a prediction model

based on genetic polymorphisms would enable to extract the group more susceptible to development of this disabling disease. As some environmental factors are already known and proven like exposure to cigarette smoke, this group could be the target of primary prevention.

In this study, we focused on analyzing a homogeneous group of ACPA+ RA patients in terms of non-HLA genetic factors that may indicate a potential risk in the development of the disease. Our proposed ML models achieved a sensitivity and specificity of approximately 70%, depending on the specific model used. In mathematical models for medical use, especially for disease risk assessment, the sensitivity and specificity of the model at the level of about 70% is satisfactory, although not perfect. There is no similar study in the literature to compare, but the criteria for the diagnosis of some rheumatic diseases have similar values of sensitivity and specificity. The best example is the current EULAR/ACR 2010 criteria for the diagnosis of RA, whose sensitivity and specificity are estimated at 73.5 and 71.4%, respectively [15]. Despite such relatively low values, they set the diagnostic standard for this common rheumatic disease.

As a golden standard according to the ACR/EULAR 2010 RA classification criteria, RF, ACPA, ESR, and CRP can be used as biomarkers for diagnosis of RA. However RF and ACPA have lack optimal sensitivity, and ESR and CRP have limited specificity [10], ACPA as the most common marker achieved 67% (sensitivity) and 95% (specificity) based on The DerSimonian–Laird random-effects method [16]. However, the ethiology of RA is an interplay between genetic, environmental factors and autoimmunity triggers and it is unlikely to bear single biomarker to diagnose or predict the disease [17]. The combination of biomarkers from various fields seems to be an adequate strategy for future analysis. ML methods can effectively handle such complex data and identify predictive patterns that may not be apparent through traditional statistical methods. Early detection of RA and identification of patients at risk for developing severe disease can allow for early intervention and more personalized treatment approaches, improving patient outcomes and reducing healthcare costs.

In the experimental part of our study, we evaluated several ML classifiers and identified the most accurate ones. We found that the selected models showed very similar levels of accuracy, but each had its unique strengths. Specifically, we found that DT is the most interpretable.

The feature importance score determined by our method has clear interpretation unlike the scores determined by comparative methods (chi2 and MRMR). It expresses an average accuracy of the classifier when a given feature is included as input, providing insight into its predictive power. Note, that the comparative methods used are filter-type methods, which evaluate features without considering the context of the applied classifier. Our proposed method is globally optimal as it evaluates all possible combinations of features in the context of a given classifier. A limitation of this approach lies in the number of features. When dealing with a large number of features, the search space can become too vast to efficiently find the optimal feature subset. In these cases, wrapper-type methods of feature selection can be more suitable. These methods include sequential forward and backward selection, genetic algorithms, or tournament searching [18].

## 5 Conclusion

Using ML for predicting RA with ACPA autoantibodies based on non-HLA gene polymorphisms would potentially enable to determine the group of individuals more prone to develop rheumatoid arthritis and further implement more precise preventive methods like smoking cessation. Our research has character and have some limitations including small size of the study group or the omission of analysis of environmental factors, such as smoking, which is a strong risk factor for RA.

Our findings suggest that ML algorithms can effectively handle complex genomic data and identify predictive patterns for RA. The identification of genetic markers for RA can have important clinical implications, including early diagnosis and personalized treatment planning. We identified the most accurate ML classifiers and found that each had its unique strengths, with decision tree being the most interpretable. Our proposed method of feature importance estimation has a clear interpretation and is globally optimal, but the limitation of the approach lies in the number of features.

Future studies will focus on incorporating additional input variables, both genetic and non-genetic, in order to improve the performance of ML models for RA prediction.

## Author Contributions

**Conceptualization:** Grzegorz Dudek, Olga Brzezińska, Joanna Sarnik, Joanna Makowska.

**Data curation:** Olga Brzezińska, Joanna Sarnik, Marta Poplawska.

**Formal analysis:** Grzegorz Dudek, Sebastian Sakowski.

**Funding acquisition:** Tomasz Poplawski, Michał Bijak.

**Investigation:** Grzegorz Dudek, Olga Brzezińska, Joanna Sarnik, Tomasz Budlewski, Grzegorz Dragan, Joanna Makowska.

**Methodology:** Grzegorz Dudek, Sebastian Sakowski.

**Project administration:** Grzegorz Dudek, Tomasz Poplawski, Joanna Makowska.

**Software:** Grzegorz Dudek, Sebastian Sakowski.

**Supervision:** Grzegorz Dudek, Tomasz Poplawski, Joanna Makowska.

**Validation:** Grzegorz Dudek, Sebastian Sakowski, Joanna Sarnik.

**Visualization:** Grzegorz Dudek, Sebastian Sakowski.

**Writing – original draft:** Grzegorz Dudek, Sebastian Sakowski, Olga Brzezińska, Tomasz Poplawski, Joanna Makowska.

**Writing – review & editing:** Grzegorz Dudek, Sebastian Sakowski, Tomasz Poplawski, Joanna Makowska.

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
