## [Decision Letter · Decision Letter 0]

17 Oct 2023

PONE-D-23-25512Machine Learning-Based Prediction of Rheumatoid Arthritis with Development of ACPA Autoantibodies in the Presence of Non-HLA Genes PolymorphismsPLOS ONE

Dear Dr. Dudek,

Thank you for submitting your manuscript to PLOS ONE. After careful consideration, we feel that it has merit but does not fully meet PLOS ONE’s publication criteria as it currently stands. Therefore, we invite you to submit a revised version of the manuscript that addresses the points raised during the review process.

**REVIEWER 1** 

1. Results are not convicing in terms of Accuracy and Sensibility and are not supported by an statistical validation with other approaches. In my opinion, is just an implementation of Matlab without any exploration of the parameters of the different algorithms.

2. The parameters of the ML algorithms should be explored for optimization.

**REVIEWER 2**

1. Why is the random forest used in the last Fig 5, but not used in the preliminary experiments (Table 4)?

2. Table 3 does not list the hyperparameters for random forest.

3. On Page 10 (line 278), is reference 13 an appropriate citation? It seems to be a paper mentioning macrophages and autoimmune diseases.

4. The paragraph on Page 10 (lines 277-284) seems unnecessary.

5. On Page 10, line 287, is there significance to the sensitivity and specificity of about 70%? How would this be utilized clinically?

**REVIEWER 3**

The Abstract should have the main findings communicated in it. The study settings should be clarified.Revise figure legends.Consider the imbalance in datasets when interpreting the models.

**REVIEWER 4**

1. In the abstract, the authors should determine which type/(s) of ML model have been included. Besides, the comparison of other methodologies or even comparison with different ML models should be mentioned with some important results.

2. The presentation and organization of the manuscript are poor.

3. The introduction is poorly written.

4. The authors should present their contribution clearly.

5. What is the motivation of the present study?

6. It is recommended to the author to add a descriptive paragraph at the end of the introduction illustrating the structure of the manuscript.

7. The sections should be numbered.

8. The authors mentioned "Machine learning (ML) and artificial intelligence (AI) have become increasingly popular tools ...". However, ML is already under the umbrella of the AI.

9. Is Genotyping an individual section or a subsection? If so, it is too short and should be merged with another section.

10. From the section called "Machine learning models". It seems the present manuscript is a comparative study. Such an important point should be clearly mentioned in the abstract and introduction. 

11. The authors mentioned that they use one or two classes. They should be consistent or as a minimum clarify on which criteria, they decide to use one or two.

12. The mentioned equations in page 8 have to be presented in a separate way with numbering.

13. It is normally to compare between all the model results together not to show each metric with some a group of them then change this group with other metric.

14. The authors are recommended to do more surveys on the related works regarding the raised crucial comments. The following paper can assist the authors in this regard: https://doi.org/10.1109/TGRS.2022.3208097;
https://doi.org/10.1109/ACCESS.2021.3076119; https://doi.org/10.1109/TGRS.2023.3296520

15. The confusion matrix should a low performance of classification.

16. How did the authors optimize the hyper-parameters.

We look forward to receiving your revised manuscript.

Kind regards,

Charalampos G Spilianakis, Ph.D

Academic Editor

PLOS ONE

Journal Requirements:

3.Thank you for stating the following financial disclosure: 

"National Science Center (NCN, Poland), grant number UMO-2017/25/B/NZ6/01358"

"This research was funded by the National Science Center (NCN, Poland), grant number 342

UMO-2017/25/B/NZ6/01358."

"National Science Center (NCN, Poland), grant number UMO-2017/25/B/NZ6/01358"

"No conflict of Interest"

6. We note that you have indicated that data from this study are available upon request. PLOS only allows data to be available upon request if there are legal or ethical restrictions on sharing data publicly. For more information on unacceptable data access restrictions, please see http://journals.plos.org/plosone/s/data-availability#loc-unacceptable-data-access-restrictions. 

Additional Editor Comments:

Based on Reviewers' Reports, either the ones positive for publication and those negative for publication, there are critical points which should be take under consideration and answered in a point by point response letter by the authors. The points which should be clarified are the following:

REVIEWER 1

1. Results are not convicing in terms of Accuracy and Sensibility and are not supported by an statistical validation with other approaches. In my opinion, is just an implementation of Matlab without any exploration of the parameters of the different algorithms.

2. The parameters of the ML algorithms should be explored for optimization.

REVIEWER 2

1. Why is the random forest used in the last Fig 5, but not used in the preliminary experiments (Table 4)?

2. Table 3 does not list the hyperparameters for random forest.

3. On Page 10 (line 278), is reference 13 an appropriate citation? It seems to be a paper mentioning macrophages and autoimmune diseases.

4. The paragraph on Page 10 (lines 277-284) seems unnecessary.

5. On Page 10, line 287, is there significance to the sensitivity and specificity of about 70%? How would this be utilized clinically?

REVIEWER 3

1. The Abstract should have the main findings communicated in it.

2. The study settings should be clarified.

3. Revise figure legends.

4. Consider the imbalance in datasets when interpreting the models.

REVIEWER 4

1. In the abstract, the authors should determine which type/(s) of ML model have been included. Besides, the comparison of other methodologies or even comparison with different ML models should be mentioned with some important results.

2. The presentation and organization of the manuscript are poor.

3. The introduction is poorly written.

4. The authors should present their contribution clearly.

5. What is the motivation of the present study?

6. It is recommended to the author to add a descriptive paragraph at the end of the introduction illustrating the structure of the manuscript.

7. The sections should be numbered.

8. The authors mentioned "Machine learning (ML) and artificial intelligence (AI) have become increasingly popular tools ...". However, ML is already under the umbrella of the AI.

9. Is Genotyping an individual section or a subsection? If so, it is too short and should be merged with another section.

10. From the section called "Machine learning models". It seems the present manuscript is a comparative study. Such an important point should be clearly mentioned in the abstract and introduction.

11. The authors mentioned that they use one or two classes. They should be consistent or as a minimum clarify on which criteria, they decide to use one or two.

12. The mentioned equations in page 8 have to be presented in a separate way with numbering.

13. It is normally to compare between all the model results together not to show each metric with some a group of them then change this group with other metric.

14. The authors are recommended to do more surveys on the related works regarding the raised crucial comments. The following paper can assist the authors in this regard: https://doi.org/10.1109/TGRS.2022.3208097;
https://doi.org/10.1109/ACCESS.2021.3076119;
https://doi.org/10.1109/TGRS.2023.3296520

15. The confusion matrix should a low performance of classification.

16. How did the authors optimize the hyper-parameters.

Reviewers' comments:

Reviewer's Responses to Questions

**Comments to the Author**

1. Is the manuscript technically sound, and do the data support the conclusions?

Reviewer #1: No

Reviewer #2: Yes

Reviewer #3: Yes

Reviewer #4: No

2. Has the statistical analysis been performed appropriately and rigorously? 

Reviewer #1: No

Reviewer #2: Yes

Reviewer #3: Yes

Reviewer #4: No

3. Have the authors made all data underlying the findings in their manuscript fully available?

Reviewer #1: No

Reviewer #2: Yes

Reviewer #3: Yes

Reviewer #4: Yes

4. Is the manuscript presented in an intelligible fashion and written in standard English?

Reviewer #1: Yes

Reviewer #2: Yes

Reviewer #3: Yes

Reviewer #4: No

5. Review Comments to the Author

Reviewer #1: This paper presents a set of Machine Learning Techniques applied to the prediction of rheumatoid arthritis in the presence of a set pf genes polymorphisms. The paper is well organized and is easy to read, with good explanations of the machine learning techniques applied in the paper. However, in my opinion, there are not significant contributions in this work, either in the computer science field, nor in the prediction techniques.

Results are not convicing in terms of Accuracy and Sensibility and a re not compared with other approaches. the parameters of the ML algorithms should be explored for optimization.

Reviewer #2: Overall, the study is rigorously executed, novel and highly clinically relevant. The experiments are performed correctly and the results are presented well without identifiable errors or significant bias. I think this machine learning algorithms can become a non-invasive clinical tool for the diagnosis of RA in the future. I have only a few suggestions/questions:

1) Why is the random forest used in the last Fig 5, but not used in the preliminary experiments (Table 4)?

2) Table 3 does not list the hyperparameters for random forest.

3) On Page 10 (line 278), is reference 13 an appropriate citation? It seems to be a paper mentioning macrophages and autoimmune diseases.

4) The paragraph on Page 10 (lines 277-284) seems unnecessary.

5) On Page 10, line 287, is there significance to the sensitivity and specificity of about 70%? How would this be utilized clinically?

Reviewer #3: The manuscript is well written. Just minor edits are required prior to publication:

Abstract should have the main findings communicated in it.

The study settings should be clarified

Revise figure legends

Consider the imbalance in datasets when interpreting the models

Reviewer #4: The manuscript has many flaws that should be carefully considered as follows:

1. In the abstract, the authors should determine which type/(s) of ML model have been included. Besides, the comparison of other methodologies or even comparison with different ML models should be mentioned with some important results.

2. The presentation and organization of the manuscript are poor.

3. The introduction is poorly written.

4. The authors should present their contribution clearly.

5. What is the motivation of the present study?

6. It is recommended to the author to add a descriptive paragraph at the end of the introduction illustrating the structure of the manuscript.

7. The sections should be numbered.

8. The authors mentioned "Machine learning (ML) and artificial intelligence (AI) have become increasingly popular tools ...". However, ML is already under the umbrella of the AI.

9. Is Genotyping an individual section or a subsection? If so, it is too short and should be merged with another section.

10. From the section called "Machine learning models". It seems the present manuscript is a comparative study. Such an important point should be clearly mentioned in the abstract and introduction.

11. The authors mentioned that they use one or two classes. They should be consistent or as a minimum clarify on which criteria, they decide to use one or two.

12. The mentioned equations in page 8 have to be presented in a separate way with numbering.

13. It is normally to compare between all the model results together not to show each metric with some a group of them then change this group with other metric.

14. The authors are recommended to do more surveys on the related works regarding the raised crucial comments. The following paper can assist the authors in this regard: https://doi.org/10.1109/TGRS.2022.3208097;
https://doi.org/10.1109/ACCESS.2021.3076119;
https://doi.org/10.1109/TGRS.2023.3296520

15. The confusion matrix should a low performance of classification.

16. How did the authors optimize the hyper-parameters.

6. PLOS authors have the option to publish the peer review history of their article (what does this mean?). If published, this will include your full peer review and any attached files.

Reviewer #1: No

Reviewer #2: No

Reviewer #3: No

Reviewer #4: No

---

## [Author Response · Author response to Decision Letter 0]

5 Jan 2024

We have included our responses to the reviewers' comments in a separate attached file.

---

## [Decision Letter · Decision Letter 1]

5 Mar 2024

Machine Learning-Based Prediction of Rheumatoid Arthritis with Development of ACPA Autoantibodies in the Presence of Non-HLA Genes Polymorphisms

PONE-D-23-25512R1

Dear Dr. Dudek,

We’re pleased to inform you that your manuscript has been judged scientifically suitable for publication and will be formally accepted for publication once it meets all outstanding technical requirements.

Kind regards,

Charalampos G Spilianakis, Ph.D

Academic Editor

PLOS ONE

Additional Editor Comments (optional):

Reviewers' comments:

Reviewer's Responses to Questions

**Comments to the Author**

1. If the authors have adequately addressed your comments raised in a previous round of review and you feel that this manuscript is now acceptable for publication, you may indicate that here to bypass the “Comments to the Author” section, enter your conflict of interest statement in the “Confidential to Editor” section, and submit your "Accept" recommendation.

Reviewer #2: All comments have been addressed

Reviewer #3: All comments have been addressed

2. Is the manuscript technically sound, and do the data support the conclusions?

Reviewer #2: Yes

Reviewer #3: Yes

3. Has the statistical analysis been performed appropriately and rigorously? 

Reviewer #2: Yes

Reviewer #3: Yes

4. Have the authors made all data underlying the findings in their manuscript fully available?

Reviewer #2: Yes

Reviewer #3: Yes

5. Is the manuscript presented in an intelligible fashion and written in standard English?

Reviewer #2: Yes

Reviewer #3: Yes

6. Review Comments to the Author

Reviewer #2: The authors have very satisfactorily addressed my questions and suggestions, including further helpful experimental data. I have no further concerns at this point.

Reviewer #3: Thanks for submitting the revised version. No further comments. All comments have been addressed in the revised version of the manuscript.

7. PLOS authors have the option to publish the peer review history of their article (what does this mean?). If published, this will include your full peer review and any attached files.

Reviewer #2: **Yes: **Yoshihiko Usui

Reviewer #3: No

---

## [Editor Report · Acceptance letter]

12 Mar 2024

PONE-D-23-25512R1 

PLOS ONE

Dear Dr. Dudek, 

I'm pleased to inform you that your manuscript has been deemed suitable for publication in PLOS ONE. Congratulations! Your manuscript is now being handed over to our production team.

Kind regards, 

on behalf of

Dr. Charalampos G Spilianakis 

Academic Editor

PLOS ONE